# Reporter Gene-Based qRT-PCR Assay for Rho-Dependent Termination In Vivo

**DOI:** 10.3390/cells12222596

**Published:** 2023-11-09

**Authors:** Monford Paul Abishek N, Heungjin Jeon, Xun Wang, Heon M. Lim

**Affiliations:** 1Department of Biological Sciences, College of Biological Sciences and Biotechnology, Chungnam National University, Daejeon 34134, Republic of Korea; monford.21692@gmail.com; 2Infection Control Convergence Research Center, Chungnam National University College of Medicine, Daejeon 34134, Republic of Korea; livinglogos@cnu.ac.kr; 3National Key Laboratory of Agricultural Microbiology, College of Life Science and Technology, Huazhong Agricultural University, Wuhan 430070, China

**Keywords:** Rho-dependent termination, reporter gene, qRT-PCR, operon

## Abstract

In bacteria, the Rho protein mediates Rho-dependent termination (RDT) by identifying a non-specific cytosine-rich Rho utilization site on the newly synthesized RNA. As a result of RDT, downstream RNA transcription is reduced. Due to the bias in reverse transcription and PCR amplification, we could not identify the RDT site by directly measuring the amount of mRNA upstream and downstream of RDT sites. To overcome this difficulty, we employed a 77 bp reporter gene *argX*, (coding tRNA^arg^) from *Brevibacterium albidum*, and we transcriptionally fused it to the sequences to be assayed. We constructed a series of plasmids by combining a segment of the galactose (*gal*) operon sequences, both with and without the RDT regions at the ends of cistrons (*galE*, *galT*, and *galM*) upstream of *argX*. The RNA polymerase will transcribe the *gal* operon sequence and *argX* unless it encounters the RDT encoded by the inserted sequence. Since the quantitative real-time PCR (qRT-PCR) method detects the steady state following mRNA synthesis and degradation, we observed that tRNA^arg^ is degraded at the same rate in these transcriptional fusion plasmids. Therefore, the amount of tRNA^arg^ can directly reflect the mRNA synthesis. Using this approach, we were able to effectively assay the RDTs and Rho-independent termination (RIT) in the *gal* operon by quantifying the relative amount of tRNA^arg^ using qRT-PCR analyses. The resultant RDT% for *galET*, g*alTK*, and at the end of *galM* were 36, 26, and 63, individually. The resultant RIT% at the end of the *gal* operon is 33%. Our findings demonstrate that combining tRNA^arg^ with qRT-PCR can directly measure RIT, RDT, or any other signal that attenuates transcription efficiencies in vivo, making it a useful tool for gene expression research.

## 1. Introduction 

Two forms of transcription termination mechanisms have been reported in bacteria [1,2]: (1) intrinsic termination or Rho-independent termination (RIT) by the terminator hairpin formed on the transcript at the end of the last transcriptional unit followed by U-tracts and (2) Rho-dependent termination (RDT). The Rho protein binds to the transcription elongation complex early in the transcription process as an elongation factor that recognizes cytosine-rich RNA sequence (Rho utilization site; rut site) in nascent mRNA and moves along the RNA chain in the 5′→3′ direction, thereby separating the RNA polymerase (RNAP) from the template DNA strand, leading to transcription termination [2,3]. In *Escherichia coli*, Rho is essential for survival, and about >20–30% of its genes are terminated by Rho, indicating its importance in *Escherichia coli* [4,5]. Although researchers have been studying the working mechanisms of Rho for decades through genetic and biochemical experiments, several issues make predicting and determining RDT regions in vivo still difficult [3,6,7], including the fact that (i) the mechanism complexity of RDT involves multiple components, including RNAP, the nascent RNA transcript, and the Rho protein. The interactions between these components make understanding the mechanism of RDT difficult; (ii) Rho is not specific to any mRNA sequence and can bind to any RNA sequence containing a rut site; (iii) the instability of Rho-terminated RNA 3′ ends: the 3′ end of Rho-terminated mRNA is rapidly digested by 3′ to 5′ exonuclease digestion after RDT due to the lack of a secondary structure.

The galactose (*gal*) operon of *Escherichia coli* has four structural genes, each of which is ~1 kb in size [1,8,9,10]. Transcription started from the promoters produces the full-length mRNA, named *galETKM*, of 4.3 kb in size [1,8,9,10]. Three mRNA species from the *gal* operon, *galE*, *galET*, and *galETKM*, have their 3′ ends within 200 nucleotides from the stop codon of the corresponding gene. Thus, the operon produces mRNAs at the end of each comprising cistron. Because the 3′ ends of these mRNAs are located at the end of each cistron, the production of these mRNAs, per se, establishes what has been referred to as natural polarity, which is more gene expression of the genes closer to the promoter [10]. Our previous research recognized three RDTs that span the *galE–galT* and *galT–galK* cistron junctions, the first and second structural genes of the galactose operon, as well as downstream of *galM*, the last structural gene of the *gal* operon [1,8,9,10].

Several issues make studying RDT difficult: Rho is not specific to any mRNA sequence and can bind to any RNA sequence containing a Rho utilization (rut) site, and the formation of stable RNA secondary structures can interfere with Rho binding to the rut site, resulting in inefficient or incomplete transcription termination. Using synthetic small RNAs (sysRNAs), we were also able to identify the in vivo RDT site downstream of galM [7]; however, we were unable to identify the RDT sites covering the *galE–galT* and *galT–galK* cistron junctions, possibly due to ribosome and sysRNA competition. Therefore, to gain a deeper insight into the RDT mechanism, it is necessary to carefully consider these issues and apply more widely applicable techniques for RDT identification.

The quantitative real-time PCR (qRT-PCR) technique is used to determine the amount of mRNA in a sample, which can be used as an indicator of gene expression [11,12]. qRT-PCR is widely used in many research areas, including gene expression profiling, validation of gene expression arrays, disease diagnosis, drug development, and forensic analysis [13,14,15]. qRT-PCR was used to validate RDT sites. Chhakchhuak and Sen designed multiple qRT-PCR primer probes and compared the gene expressions of Rho function-deficient strains relative to the wild-type strains, and the upregulation of the genes implied the presence of RDTs upstream of these probe sites [16]. Notably, this approach does not reflect changes in the relative amount of transcripts upstream and downstream of RDT sites; it reflects changes in the relative amounts of gene expression between the wild type and Rho function-deficient strains. Due to bias in reverse transcription and PCR amplification, the quantities between two different transcripts cannot be directly compared [17,18]. Therefore, researchers cannot directly compare the difference in transcript amounts between the upstream and downstream RDT sites.

We wondered if there is a way to fairly show the amount of transcription between the upstream and downstream of RDT sites, which would more directly show transcription termination and could also be more easily used to calculate transcription termination efficiency. In this study, we employed a transcriptional reporter system, tRNA^arg^ from *Brevibacterium albidum*, which was transcriptionally fused to the sequences to be assayed. With this approach, transcription initiated from the *gal* operon promoters transcribes the inserted *gal* sequence, tRNA^arg^, unless it is terminated by an upstream transcription termination that is intended to be assayed. We assayed the RDT region by quantifying the relative amount of tRNA^arg^ using qRT-PCR analyses. With this approach, we observed a gradual decrease in *gal* operon transcripts due to RDT, and we were also able to calculate the efficiency of RDT. This approach represents a valuable tool for advancing future investigations into RDT and gene expression.

## 2. Materials and Methods

### 2.1. Bacterial Strains, Media, and Growth Conditions

For RDT analysis, *Escherichia coli* MG1655, HME60 (*rho-*15) (provided by D. Court, NIH, MD, USA) [7,10,19], and MG1655 Δ*spf* [8,20] strains were used. Chromosomal deletion strains of *Escherichia coli* MG1655 were generated using phage Lambda Red-mediated recombineering [21]. For plasmid construction, the DH5α strain was used. Primers used in this study are listed in Appendix A. All cells were grown at 37 °C in Lysogeny Broth supplemented with 0.5% (*w*/*v*) galactose and either chloramphenicol (15 μg/mL) or ampicillin (100 μg/mL) to reach an exponential growth phase with an optical density at 600 nm (OD_600_) of 0.6. Following that, RNA isolation was performed.

### 2.2. Design and Construction of Plasmids

Sequences of *argX* from *Brevibacterium albidum* corresponding to tRNA^arg^ were amplified by PCR using primer sequences shown in Appendix A. The PCR fragment was digested with *Hin*d III and then inserted immediately upstream of a promoterless chloramphenicol resistance gene (*cat*) gene in a medium-copy-number plasmid pKK232-8 (GE Healthcare, OH, USA) to create the pHL1141 plasmid [10,22] (Figure 1, Appendix A). To assay RDT, the *gal* operon DNA sequence was cloned between the *Bam*HI and *Sal*I restriction enzyme sites upstream of *argX* in pHL1141.

To generate plasmids pHL1031, pHL1193, pHL2000, pHL2184, pHL4270, pHL4344, and pHL4444, the *gal* operon DNA sequences from *gal* coordinates −73 to 1031, −73 to 1193, −73 to 2000, −73 to 2184, −73 to 4270, −73 to 4333, and −73 to 4444 were obtained with the corresponding primer pairs via PCR amplification (Appendix A). The resulting PCR fragments were ligated to pHL1141, which has been digested with *Bam*HI and *Sal*I (Figure 1). DNA sequencing was used to confirm the plasmid constructs.

### 2.3. RNA Preparation and Reverse Transcription-Quantitative PCR (qRT-PCR)

Following the manufacturer’s instructions, the clarified cell lysates (2 × 10^8^ cells) were used to purify total RNA using the Direct-zol RNA MiniPrep kit (Zymo Research, CA, USA). After Turbo DNase I (Thermo Fisher Scientific, Rockford, IL, USA) reaction to remove genomic or plasmid DNA, one microgram of total RNA was reverse-transcribed in a 20 μL reaction volume as previously described [10]. Under the following conditions, qRT-PCR was carried out in 10 μL reaction mixtures containing 5 μL of iQ SYBR green Supermix (Bio-Rad, Hercules, CA, USA), 3 μL of nuclease-free water, 0.5 μL each of forward and reverse primer (10 μM) (Appendix A), and 1 μL of the cDNA template. The qRT-PCR conditions were as follows: 94 °C for 3 min (initial denaturation), and then 40 cycles of 15 s of denaturation at 94 °C, 20 s of hybridization at 60 °C, and 15 s of elongation at 72 °C (CFX96; Bio-Rad, Hercules, CA, USA). The results from each sample were normalized using the internal control *rrsB*, which encodes 16S rRNA. The qRT-PCR reactions were set up in triplicates, and the mean Ct value was used to analyze three independent experiments further. The expression of each sample is presented as mean ± SD [23,24,25]. The ΔΔCT values were subjected to a one-way analysis of variance (ANOVA) using the Bonferroni test.

### 2.4. RNA Stability Assay

We examined the decay rate of *galE*, tRNA^arg^, 16S rRNA, and *cat* mRNA species using wild-type (WT) MG1655 cells harboring plasmids [10] (as detailed in Table 1 and Table 2). We added 100 μg/mL of rifampicin, a transcription inhibitor (38), to LB-grown cells with an OD600 of 0.6 to determine RNA stability. Total RNA was extracted for qRT-PCR analysis after samples were obtained at 0, 2, 4, and 8 min after rifampicin addition. We used 16S rRNA to standardize the findings. The relative expression was calculated using the 2−ΔΔCT method by averaging the fold changes of three replicates from three independent experiments.

### 2.5. Quantification and Statistical Analysis

Each experiment was independently repeated at least three times with similar results. Data were subjected to one-way analysis of variance (ANOVA) using the Bonferroni test, *n* = 3. GraphPad Prism version 9.0 (2021) was used for one-way ANOVA and graph plotting.

## 3. Results

### 3.1. Establishment of RDT In Vivo Detection System

We assayed RDTs using the pBR322-derived plasmid pKK232-8 for RDT detection (Figure 1, Appendix A). pKK232-8 has the advantage of the presence of multiple *rrnB-*strong RIT terminators upstream and downstream of the *cat* gene avoiding interference with upstream and downstream transcriptional read-through. We chose tRNA^arg^ from *Brevibacterium albidum* as a reporter because previous studies have shown that its sequence shares little homology with tRNAs in *Escherichia coli* and is quite stable [22,26]. First, we quantified tRNA^arg^ in MG1655-pKK232, where the *argX* gene was absent by qRT-PCR, and the data showed that the C(t) value was greater than 35, indicating that it could not be detected. Then, we inserted the 77 bp gene *argX* coding tRNA^arg^ upstream *cat*, quantified the amount, and measured the half-life of tRNA^arg^ in the MG1655-pHL1141 strain by qRT-PCR. The C(t) values increased to 13.1, indicating that tRNA^arg^ was successfully transcribed. Contrary to our expectation, the tRNA^arg^ failed to show strong stability; instead, it had a similar half-life to *gal* mRNA, about 1.2 min (Table 1). We also tested the half-life of *cat* mRNA, as well as the half-life of 16S rRNA, and the results showed that the half-life of *cat* mRNA was about 1.2 min and 16S rRNA was very stable, with no degradation observed (Table 1).

We then fused and inserted various *gal* operon sequences upstream of *argX* to construct seven pHL1141-derived plasmids (Figure 1) and transformed them in *Escherichia coli* MG1655 strains as described in Materials and Methods. We tested the half-lives of tRNA^arg^ expressed in these strains. The results showed that tRNA^arg^ half-lives were similar, ranging from 1.0 to 1.3 min (Table 2). Since the amount of RNA quantified by qRT-PCR is the steady state after RNA synthesis and degradation, and the degradation rates of tRNA^arg^ on these plasmids are similar, we believe that the amount of tRNA^arg^ from these strains would represent the mRNA synthesis before and after the putative RDT site, respectively.

### 3.2. RDT at the galE–galT Cistron Junction

We measured the efficiency of RDT by measuring the amount of the *gal* transcript RNA before and after the RDT event at 1183 in vivo [9]. To measure the transcription ‘before-RDT,’ the portion of the *gal* operon was from −73 to 1031. To measure the transcription ‘after-RDT,’ the portion was from −73 to 1193. The resultant plasmids were pHL1031 and pHL1193, respectively (Figure 1). These regions of *gal* contain all *cis-*acting elements necessary for *gal* expression. Amounts of tRNA^arg^ from pHL1031 and pHL1193 would represent the amounts of *gal* transcript initiated from the *gal* promoters before and after the RDT at 1183, respectively. We measured tRNA^arg^ expression in these two strains harboring pHL1031 and pHL1193 plasmids individually. The C(t) values were 12.8 and 13.3, indicating that tRNA^arg^ was successfully transcribed. The results indicated that in strain MG1655-pHL1193, tRNA^arg^ was 64 ± 19% of the level found in MG1655-pHL1031 (Figure 2A). This suggests that transcription termination occurs approximately 36% of the time at 1183 due to Rho in MG1655 (Table 3). In HME60 (*rho-*15) cells, where Rho is non-functional [7,10,19] and has a C-terminus that is nine amino acids different from the WT’s fused with the *bla* (Amp^R^) gene (D. Court, NIH, USA), the tRNA^arg^ expression level was almost the same as that in MG1655-pHL1031 (Figure 2B), which confirms that the measurement was of RDT at the *galE*–*galT* junction.

### 3.3. RDT at the galT–galK Cistron Junction and Spot 42 Enhanced It

We measured the efficiency of RDT that results in the generation of the mRNA *galET-short* at *2121–2125* [8]. To measure transcription, a portion of the *gal* operon from −73 to 2000 was cloned to generate the plasmid pHL2000. Another portion from −73 to 2184 was cloned to generate the plasmid pHL2184 (Figure 1). Since the in vitro assay demonstrated that Rho terminates transcription at 2184 [8,27], we anticipated that the amount of tRNA^arg^ would be less from MG1655-pHL2184 than MG1655-pHL2000. The C(t) values were 14.3 and 14.6, indicating that tRNA^arg^ was successfully transcribed. The result, indeed, showed that tRNA^arg^ in MG1655-pHL2184 is 74 ± 19% of that in MG1655-pHL2000 (Figure 3A), suggesting that Rho terminates transcription about 26% of the time in the presence of the RDT site.

Spot 42, a 109 nucleotide-long noncoding sRNA, binds to the middle of multi-cistronic mRNA at the *galT*–*galK* cistron junction and enhances RDT [20]. Our qRT-PCR results showed that in the Δ*spf*-pHL2184 strain in which the Spot 42 coding gene was deleted, tRNA^arg^ after the RDT is 92 ± 10% of that of Δ*spf*-pHL2000 (Figure 3B), suggesting that in the absence of Spot 42, Rho hardly terminates transcription even when the RDT site is present. These results demonstrate that Spot 42 is the critical factor that causes RDT to generate the *galET* mRNA. We also observed that the tRNA^arg^ in HME60-pHL2184 and HME60-pHL2000 were almost equal, regardless of the presence of RDT sequences (Figure 3C). The above findings with various strains demonstrated that measuring RDT efficiency with the qRT-PCR technique is quite effective.

### 3.4. RDT and RIT at the End of the Operon

At the end of the *gal* operon, transcriptions go through two terminators: the upstream RIT and downstream RDT [1]. Rho terminates transcripts at 4409 and has been quickly processed by exonuclease digestion [1]. For measurement of transcription, ‘before-*RIT*-*RDT*’ (from −73 to 4270), and transcription, ‘after-*RIT’* and ‘after-*RIT*-*RDT’* (−73 to 4344 and −73 to 4444). The resultant plasmids were pHL4270, pHL4344, and pHL4444, respectively (Figure 1). The half-lives of tRNA^arg^ expressed by MG1655-pHL4270, MG1655-pHL4344, and MG1655-pHL4444 strains were similar, about 1.3, 1.2, and 1.3 min, respectively (Table 1). Thus, the amount of tRNA^arg^ from MG1655-pHL4270, MG1655-pHL4344, and MG1655-pHL4444 would represent the amount of *gal* transcript initiated from the *gal* promoters before and after the RIT and RDT sites, respectively. The RIT and RDT sites are at 4315 and 4409, and we expected the amount of tRNA^arg^ would be less from the plasmid containing the RIT and RDT sites than from the plasmid without it.

The amount of tRNA^arg^ was measured using qRT-PCR in MG1655-derivative cells harboring the plasmids, measuring C(t) values to be 13.1, 13.3, and 14.9, indicating that tRNA^arg^ was successfully transcribed. The results showed that tRNA^arg^ in MG1655-pHL4344 was 67% in MG1655-pHL4270 (Figure 4A), suggesting an RIT efficiency of 33%, and the tRNA^arg^ in MG1655-pHL4444 was 37% in MG1655-pHL4344, which indicated an RDT efficiency of 63% (Table 3). In HME60 (*rho*-15) cells, the results showed that the tRNA^arg^ in HME60-pHL4344 is 68% of that in HME60-pHL4270 (Figure 4B). This suggests that RIT efficiency is 32%, almost the same as in MG1655-pHL4333. The expression level of tRNA^arg^ in HME60-pHL4444 was not significantly different from that in HME60-pHL4344, suggesting that the RDT no longer exerted a function in HME60 (Figure 4B).

## 4. Discussion

In the present work, our results show that qRT-PCR analysis of tRNA^arg^ can directly reflect the amount of transcript before and after the RDT site, respectively. If RDT is effective, mRNA transcript levels should decrease after termination, indicating that the Rho protein has caused transcription termination [16,28]. Additional controls can be included, such as measuring transcript levels in cells that fail to express Rho protein (here, HME60 (*rho*-15)) or have mutations in the Rho-binding site. Moreover, qRT-PCR can be a powerful tool for studying the role of the Rho protein in bacterial transcription termination, providing a quantitative measure of mRNA transcript levels before and after RDT [16,28].

Transfer RNAs (tRNAs) are produced as precursors and then go through a multi-step maturation process to become mature tRNAs with a cloverleaf-shaped secondary structure [29,30]. tRNAs create complexes with proteins known as aminoacyl-tRNA synthetases, which are in charge of attaching the correct amino acid to the correct tRNA molecule [29,30]. These protein–-tRNA complexes shield the tRNA from degradation and keep the tRNA molecule stable, making it suitable to be utilized as a reporter gene to monitor gene expression. The tRNAs are seasonably stable due to their unique secondary structure and the protective effect of the proteins bound to them [31]. tRNA can detect smaller changes in gene expression levels and does not require an exogenous substrate. As a result, prior studies utilized them as stable RNAs for reporter genes. However, in this research, we found that 77 bp tRNA^arg^ was not as stable as expected. This could be because tRNA^arg^ is a segment of a long transcript rather than a processed cloverleaf-shaped structure. As a result, it loses its initial secondary structure and physiological function, and it no longer has the reported stability.

Using tRNA^arg^ as a reporter gene to measure transcription has various advantages: (i) First and foremost, it avoids the bias inherent in translation. In gene expression studies, the RDT to be determined is typically fused with a gene that produces a protein, such as β-galactosidase or fluorescent proteins, and the intensity of the proteins serves as a measure of the transcriptional activity [32,33]. However, it has been shown that due to variations in translation rates, the quantity of transcription does not always correlate linearly with the amount of translation [34,35,36]. Our findings show that tRNA^arg^ has similar half-lives across different plasmids, which explains why the tRNA^arg^ reporter precisely reflects the combination of transcription and decay. (ii) To quantify gene expression, the well-known β-galactosidase (*lacZ*) technique requires the addition of a substrate (such as X-gal), which can introduce unpredictability and be toxic to some cell types [37]. In contrast, qRT-PCR assessment of tRNA^arg^ can detect smaller changes in gene expression levels and does not require an exogenous substrate, allowing for accurate and precise quantification of the effects of Rho-termination on gene expression.

qRT-PCR is a powerful tool for studying RDT in bacteria; still, researchers should be aware of its limitations and select the best method suited to their experimental design and research question. qRT-PCR amplifies cDNA, and amplification efficiency varies depending on the length and GC content; it also relies on the specificity of the primers used to amplify the cDNA. Mnon-specific binding or amplification can occur, resulting in false positives or inaccurate results. On the other hand, the selection of appropriate internal control is critical, and some commonly used controls may not be suitable for all experimental conditions. Although traditional approaches to studying RDT, such as Northern blotting and primer extension assays, allow for the identification of transcriptional terminators at single nucleotide resolution, they are time-consuming and, in particular, rely on the handling of hazardous ^32^P-labeled primers and probes radioisotopes for radioactivity detection [1,7,8,9,10,27,38]. These methods may also necessitate more RNA and are unsuitable for high-throughput analysis. qRT-PCR is, thus, a robust and efficient technique for studying RDT in bacteria, and its benefits make it an appealing option for researchers in this field.

## 5. Conclusions

Rho is a protein that is crucial for bacterial transcription termination. It plays an important role in the release of mRNA by binding to target sites on nascent RNA and causing transcription termination. The difficult molecular mechanisms and real-time nature of transcription termination make studying Rho-dependent termination (RDT) in vivo hard. Here, we establish and validate the significance and utility of using tRNA^arg^ as a reporter gene to detect Rho-dependent termination (RDT) in vivo. This method, which looks at RNA expression and reflects only the transcription of the gene to which it is fused, is more direct than employing β-galactosidase or fluorescent proteins as reporter genes and should be promoted and widely used to detect transcription termination. Thus, this method offers a reliable and quantitative method for examining the role of Rho and other factors involved in transcription termination.

## Figures and Tables

**Figure 1 cells-12-02596-f001:**
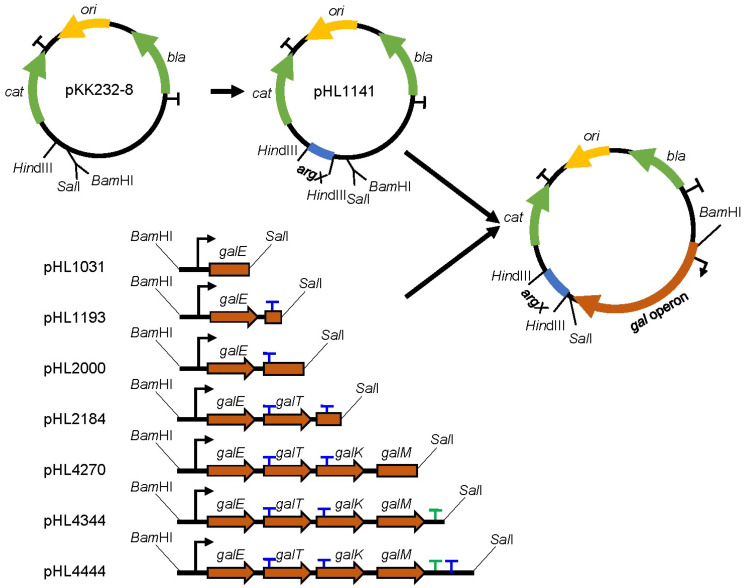
Plasmids used to measure Rho-dependent termination (RDT) efficiency in vivo and their construction strategy. The plasmid map of pKK232-8; *ori*, plasmid origin of replication (yellow); Genes: *cat* and *bla*, Chloramphenicol acetyltransferase and β-lactamase (green) antibiotic resistance genes; transcription terminators are denoted as a (T), *rrnBT1/T2* RIT terminators (black), *gal* RDT (blue), *gal* Rho-independent termination (RIT) (green); and the restriction enzymes *Hin*dIII, *Sal*I, and *Bam*HI are indicated. Sequences of *argX* corresponding to tRNA^arg^ inserted upstream of the *cat* gene are shown in blue to generate pHL1141. Schematic presentation of the DNA fragments of *gal* operon harboring from −73 to 1031, −73 to 1193, −73 to 2000, −73 to 2184, −73 to 4270, −73 to 4344, and −73 to 4444 were cloned in front of the tRNA^arg^ gene of pHL1141. The RDT site in the *gal* DNA fragments is represented by (T) at the end of the *gal* cistrons. The relative amounts of tRNA^arg^ from cells harboring pHL1031 (before-RDT), pHL2000 (before-RDT), pHL4270 (before-RIT/RDT), pHL1193 (after-RDT), pHL2184 (after-RDT), pHL4344 (after-RIT), and pHL4444 (after-RIT/RDT) were taken to represent the efficiency of RIT and RDT.

**Figure 2 cells-12-02596-f002:**
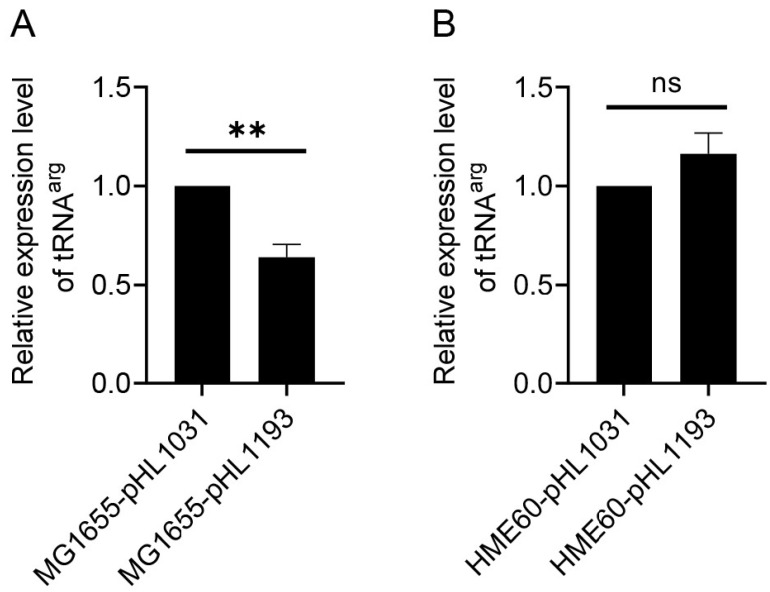
RDT at the *galE–galT* cistron junction. The results of quantitative real-time PCR (qRT-PCR) measurement of tRNA^arg^ in (**A**) MG1655 and (**B**) HME60 (*rho-*15) are presented in which the relative amount of tRNA^arg^ in pHL1193 (after-*RDT*) is compared to that in pHL1031 (before-*RDT*). Error bars represent the mean fold-change ± standard deviation of three replicates from three independent experiments (n = 3). ns: not significant, *p* value > 0.05; **: significant differences, 0.001 < *p* value  < 0.01.

**Figure 3 cells-12-02596-f003:**
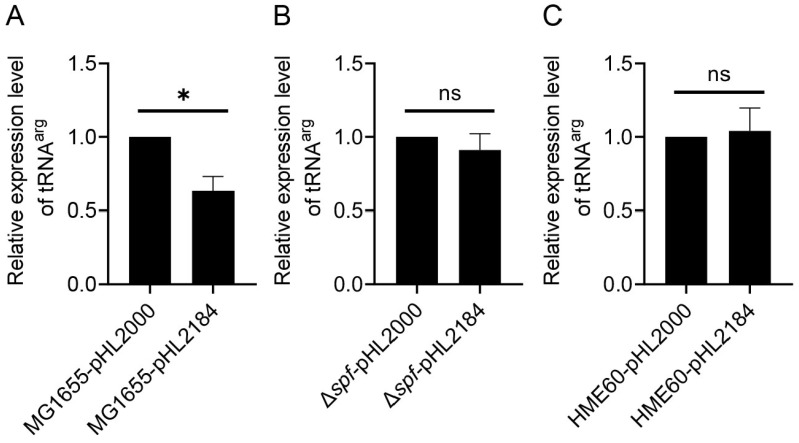
Spot 42 controls RDT at the *galT–galK* cistron junction. The results of qRT-PCR measurement of tRNA^arg^ in (**A**) MG1655, (**B**) MG1655Δ*spf*, and (**C**) HME60 (*rho-*15) are presented in which the relative amount of tRNA^arg^ in pHL2184 (after-*RDT*) is compared to that in pHL2000 (before-*RDT*). Error bars represent the mean fold-change ± standard deviation of three replicates from three independent experiments (n = 3). ns: not significant, *p* value > 0.05; *: significant differences, 0.01 < *p* value  <  0.05.

**Figure 4 cells-12-02596-f004:**
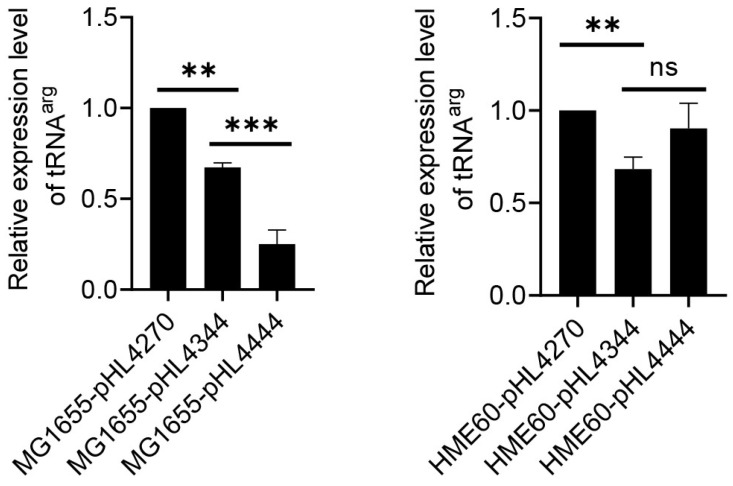
RDT at the end of *gal* operon. The results of qRT-PCR measurement of tRNA^arg^ in (**A**) MG1655 and (**B**) HME60 (*rho-*15) are presented in which the relative amount of tRNA^arg^ in pHL4344 (after-*RIT*)/pHL4444 (after-*RDT*) is compared to that in pHL4270 (before-*RIT/RDT*). Error bars represent the mean fold-change ± standard deviation of three replicates from three independent experiments (n = 3). ns: not significant, *p* value >0.05; **: significant differences, 0.001 < *p* value  <  0.01. ***: significant differences, *p* < 0.001.

**Table 1 cells-12-02596-t001:** Various RNA half-lives.

RNA Name	Half-Life (min)
tRNA^arg^	1.2 ± 0.09
*galE*	1.1 ± 0.17
*cat*	1.5 ± 0.14
16S rRNA	Stable (>4000)

**Table 2 cells-12-02596-t002:** tRNA^arg^ half-lives.

RNA Name	From Plasmid	Half-Life (min)
tRNA^arg^	pHL1031	1.2 ± 0.10
	pHL1193	1.0 ± 0.08
	pHL2000	1.3 ± 0.11
	pHL2184	1.3 ± 0.09
	pHL4270	1.3 ± 0.17
	pHL4344	1.2 ± 0.12
	pHL4444	1.3 ± 0.08

**Table 3 cells-12-02596-t003:** Termination efficiency calculated. The termination efficiency % was calculated using the formula: Termination frequency = 1 − read-through/upstream transcripts.

Termination Efficiency	
E-T junction RDT	36%
T-K junction RDT	26%
*galM* end RIT	33%
*galM* end RDT	63%

## Data Availability

All the data used to support the conclusions are available in the research article and the Appendix A.

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
