# Peer review of "Reporter Gene-Based qRT-PCR Assay for Rho-Dependent Termination In Vivo"

_cells, 2023, doi:10.3390/cells12222596_

Round 1

Reviewer 1 Report

Comments and Suggestions for Authors

I have reviewed the manuscript by Abishek N et al. It describes a method to detect and quantify the amount of transcription of gene in vivo using qTR-PCR and a reporter gene (argX transfer RNA of B. albidium). In fact, the technic can also quantify the amount of transcription up- and downstream of a signal e.g., transcription terminators, the authors have shown.

The manuscript, if published, may be useful source of gene transcription analysis in a more unbiased way than using other reporter systems as the authors claim, and I agree. Thus, I recommend publication in Cells but after several issues mentioned below have been clarified by the authors: \

1.       To my knowledge, the rho gene is essential in E. coli. The rho defective mutant the authors have isolated and used in this paper is described at one place as rho15 (line 89) and at another place as rho::Ampr (line 190), which means that the AmpR gene has been inserted within the rho gene.

The authors also mention elsewhere that the rho defective mutant they have used are completely devoid of Rho function. Is this true? The authors have referred to several of their own publications, but a reader needs to know now the answer to these questions here in this Athe rho function, and why the strain survives?

2.       In figure 2B results, why the amount of transcription is higher (it seems significantly) in strain 1193 than in 1031?

3.       Is there no RDT in the galK-galM junction?

4.       Why in figure 4 B results, it shows that in the absence of rho the transcription becomes almost 1.0. Does it mean that in the absence of rho transcription does not sdtop at the rho-independent termination signal? This is a serious issue and must be discussed.

5.       The system the authors describe does not necessarily measure RDT but can also quantify Rho-independent   or any other signal that attenuates transcription. This point should be mentioned in the Abstract.

Reviewer 2 Report

Comments and Suggestions for Authors

In this study the Authors developed a very good protocol to estimate the in vivo  efficiency of transcriptional termination  at RDT sites.

The procedure, based on RT-qPCR, is simple and  quantitative.  They used the tRNAarg  as report gene  thus preventing possible drawbacks related to translation of reporter genes coding for enzymes. The weakest and time-consuming  step of this system is the preparation of the constructs containing the genes to be tested.  Importantly, results are also supported by measuring the activity of RDTs in null mutants (rho, spf).

Minor points

Is there a particular reason for the use of pKK232-8 , harboring the  promoter-less cat gene? Might other vectors  be used?

The sentence at lines 99-101 is confusing, a  careless reader could understand that cloning has been done in pBR.

The conditions of qPCR should be reported, in particular the PCR cycle and  the method for signal detection.

The Ct of the original pKK 232 pasmid was about 35 (lines 148-151).  What Ct range has been obtained for constructs of Fig. 1.

It is quite questionable to state that primer extension and N. blotting methods are less sensitive and quantitative than qPCR (lines 288-230), in particular when 32P-labeled primers/probes are used and detection of radioactivity is carried out by imager. In addition, as the Authors very well know, primer extension permits to identify, at single nucleotide resolution, transcriptional start points and terminators. This point should be discussed.
